# ROYAL SOCIETY
# OPEN SCIENCE

environmental science/complexity

mobile phone data, data science, call detail records

**Author for correspondence:**
Federico Botta
e-mail: f.botta@exeter.ac.uk

# Quantifying the differences in *call detail records*

## Federico Botta

Department of Computer Science, University of Exeter, Exeter, UK

FB, 0000-0002-5681-4535

The increasing availability of mobile phone data has attracted the attention of several researchers interested in studying our collective behaviour. Our interactions with the phone network can take several forms, from SMS messages to phone calls and data usage. Typically, mobile phone data are released to researchers in the form of *call detail records*, which contain records of different types of interactions, and can be used to analyse various aspects of our behaviour. However, the inherently behavioural nature of these interactions may result in differences between how we make phone calls and receive text messages. Studies which rely on data derived from these interactions, therefore, need to carefully consider these differences. Here, we aim to investigate differences and limitations of different types of mobile phone interactions data by analysing a large mobile phone dataset. We study the relationship between different types of interactions and show how it changes over time. We anticipate our findings to be of interest to all researchers working in the area of computational social science.

## 1. Introduction

Recent years have witnessed a surge in the use of new forms of data to study our society [1–4]. For instance, our interactions with the Internet and social media platforms have allowed researchers to better understand several aspects of our collective behaviour [5–14]. However, an even more popular data source, which has quickly become fundamental in the study of human behaviour, is that derived from our interactions with the mobile phone network [15–18]. Indeed, such data have become a key part in the new field of computational social science and has recently been of great importance in studying the current COVID-19 pandemic [19–24].

Mobile phone data are often made available to researchers in the form of *call detail records (CDR)* by mobile phone providers. These records typically contain interactions in the form of either SMSs, calls or Internet usage, alongside a time stamp and location of when and where the interaction took place. For privacy reasons, the data are often released at an aggregate

level, both temporally and spatially. However, a key aspect to consider is that mobile phone data is inherently dependent on our behaviour. As our behaviour changes, we may stop sending SMSs and start making more phone calls, or use services via the Internet such as instant messaging apps. These changes in behaviour would be reflected in CDR data which are made available to researchers. It may also be the case that pricing plans may favour the use of one form of communication over the other, thus encouraging subset of users to either make more phone calls or send more SMSs. Finally, biases in the population generating the data, both in terms of different cultures as well as biases in the population of users of different mobile phone providers, may also contribute to observed behavioural differences in CDR data. If there are differences between how we use our phones, it is clear that researchers should be aware of the limitations of using mobile phone data which have been derived, for instance, only from text messaging activity. Indeed, existing studies in the literature often rely, due to data availability, on only one form of CDR data [25,26]. Additionally, there is still no conclusive answer to the issue of how different CDR data, such as those derived from SMS or calls, can be incorporated into one simple measure, since there are inherent biases in each form of communication [16,27]. Models based on new forms of data have already shown the importance of training and calibrating [28,29], so a better understanding on the limitations of mobile phone data is crucial for researchers and stakeholders alike.

Indeed, differences in interactions generated by our use of mobile phones have already been found in previous studies [30]. Differences and similarities in SMS and phone call data have been studied using a multi-layered network approach to show that mobile phone data may be better modelled by a network of communication channels [31]. Additionally, communications via mobile phones tend to happen between people who are more likely to be near each other [31], suggesting the existence of strong spatial interactions in mobile phone data. Ego-networks of mobile phone interactions have been quantified in terms of *social signatures*, which is a way of measuring how each individual allocates their interactions across the network [32]. Social signatures have been demonstrated to be consistent at different levels, even though the interactions via different communication channels only show partial overlap for ego-networks [32]. Interactions via SMS and phone calls exhibit an interesting interplay between similarities and differences. For instance, the analysis of a massive mobile phone dataset has shown that people who make a lot of calls also send a lot of text messages [33]. However, the same result does not hold when considering incoming calls and SMSs.

Our study extends previous results in two main aspects. Firstly, we explicitly consider Internet activity as one of the different mobile phone data channels in our analysis, an aspect which has been less investigated in the literature. Internet usage on mobile phones has become increasingly important and our analysis explicitly focuses on similarities and differences between Internet interactions and regular mobile phone communications. Secondly, in our analysis, we do not rely on individual-level data to construct ego-networks, but we show that, even at the aggregate level, both temporally and spatially, mobile phone interactions exhibit interesting similarities and differences. More broadly, while it is known that there are different behaviours when making phone calls, sending text messages, or using the Internet, this study aims to quantify exactly to what extent those differences give rise to differences in the observed CDRs.

## 2. Data

We retrieve data on mobile phone activity recorded in the city of Milan, and its surroundings, during a time period of two months between 1 November 2013 and 31 December 2013 [34,35]. The dataset contains information on mobile phone activity of *Telecom Italia* users at 10 min granularity for cells in a discrete grip superimposed on the area under analysis (figure 1a). The grid contains a total of 10 000 cells, and all cells are included in our analysis presented below. The mobile phone activity is composed of CDRs, which contain information on the SMSs sent and received, incoming and outgoing calls, and usage of the Internet via mobile phones. Internet activity generates a CDR in each of the following three circumstances: the user starts an Internet connection; the user ends an Internet connection; during one connection, either 15 min elapse from the last generated CDR, or 5 MB of data have been transferred since the last generated CDR. We refer to these different types of CDRs as *CDR layers*. Further to the spatio-temporal aggregation of the CDRs, the data have also been rescaled with an unknown factor to preserve privacy by *Telecom Italia*. Figure 1b depicts the aggregated activity of each CDR for the first day of the period under analysis (1 November 2013). Initial visual inspection

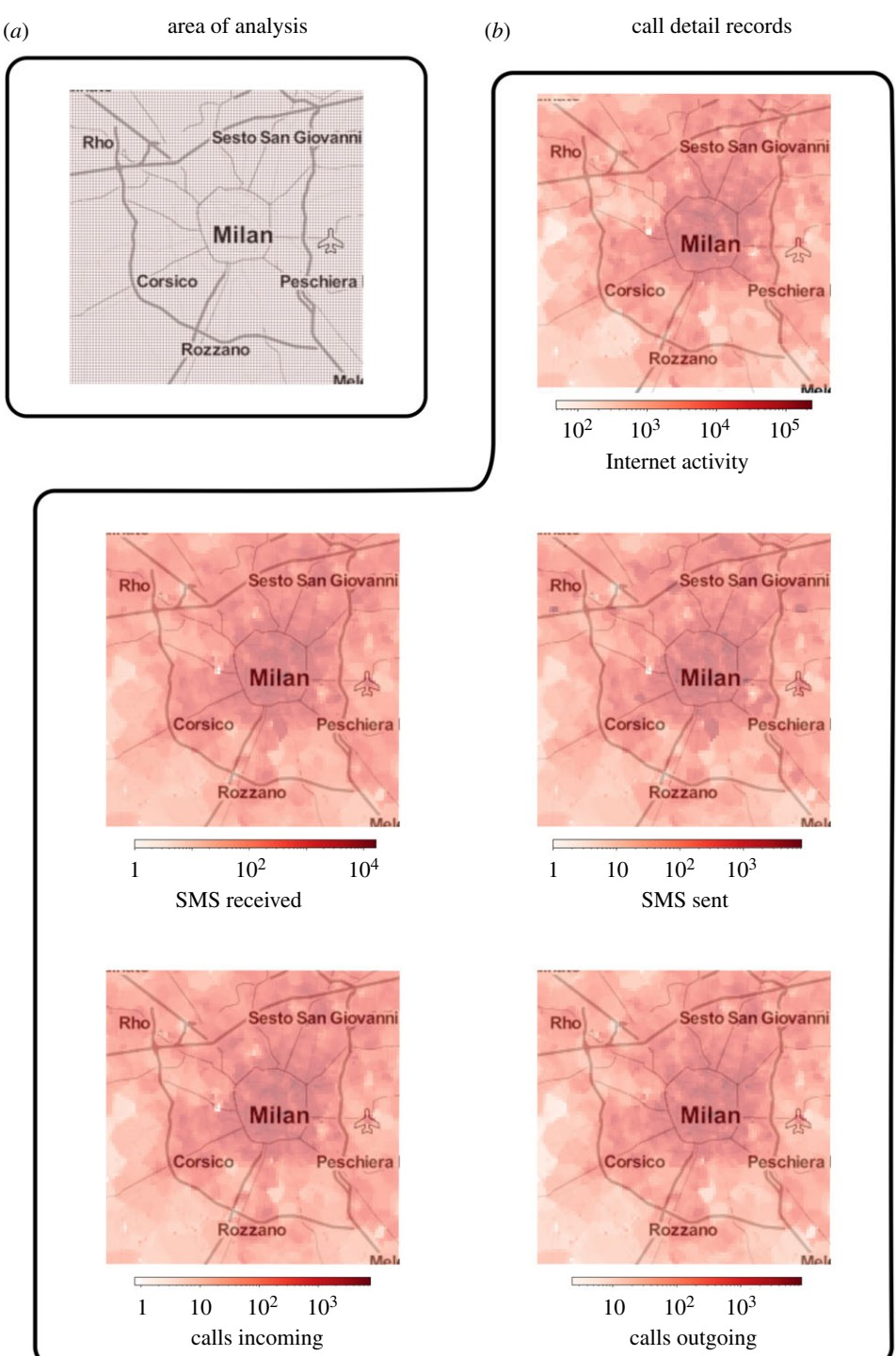

**Figure 1.** Call detail records in the city of Milan. We retrieve data on mobile phone activity of *Telecom Italia* users for the city of Milan and its surroundings during the time period between 1 November 2013 until 31 December 2013. (*a*) Here, we represent the geographical area for which the datasets are available, and where our analysis focuses. The area under analysis is divided in cells in a discrete grid superimposed on the area of Milan. This grid has 10 000 cells of size 235 × 235 m, and is presented here. This map, as all others in the figure, was created using data from *OpenStreetMap*. (*b*) We depict here the *call detail records (CDRs)* used in our analysis aggregated over one day in the period of analysis (1 November 2013). CDRs contain information on interactions via the mobile phone network, such as SMSs, calls and usage of the Internet via mobile phones. We aim to measure and compare the similarity of information contained by each of these data sources. Visual inspection indicates a broad similarity between the data sources at the aggregate level. For visualization purposes, the activity maps display $\log(x + 1)$, where $x$ is each data source, such as Internet, SMS received and so on. Finally, note that the CDRs values have been rescaled by *Telecom Italia* with an unknown factor for privacy reasons, so the activity values do not correspond to, for instance, the actual number of calls made or received.

**Table 1.** Time-series distance measures. To measure the similarity of the time series of each CDR layer in each cell, we use a range of distance measures described in [38], and briefly summarized here. All three measures are model-free approaches, meaning that they do not assume a specific time-series model to compare their similarity.

| distance measure | brief description |
| --- | --- |
| Kendall's correlation | Kendall's correlation coefficient |
| periodogram | euclidean distance between periodogram ordinates |
| discrete wavelet transform | distance between wavelet approximations of the time series |

suggests that there is a broad similarity between CDRs at the aggregate level. Data to generate the maps in figure 1 have been retrieved from *OpenStreetMap* [36].

# 3. Methods

We aim to study the relationship between different types of CDRs to investigate whether they contain similar information both across space and time. To do so, we rely on a variety of measures which allow us to compare CDRs. We use both hourly and daily aggregate data in order to compare whether the level of aggregation of the data also plays a role in the similarity between types of CDRs. When aggregating and analysing the data, it is important to highlight that in our analysis we do not differentiate between working days and weekends. For each level of temporal aggregation, we test whether cells with a higher activity in one CDR layer correspond to higher activity in a different CDR layer. We test this using *Kendall's correlation coefficient*, which compares the rank of one CDR layer with the ranks of another layer. Since we repeat this for every hourly time interval, we correct for multiple hypothesis testing by adjusting the $p$-values using the *false discovery rate* (*fdr*) correction [37]. We also perform a simple linear regression between the CDR layers, and analyse the evolution of the regression coefficient. However, it is important to highlight that this analysis is not intended to look for any causal relationship between the variables, rather it only shows how the relationship between the CDR layers evolves over time.

We then focus on the differences at the individual cell level. To do so, we first construct an hourly and a daily standardized time series for each cell and each CDR layer. We then calculate a range of distance measures between these time series with the idea of comparing the similarity between different CDR layer in each cell. The measures we use are reported and briefly described in table 1. A more complete description of them can be found in [38].

# 4. Results

As a preliminary analysis, we aggregate the data in each cell over the entire period of analysis. Each cell, therefore, has five values corresponding to the five CDR layers under analysis. We then calculate Kendall's correlation coefficient for each pair of layers and we present the results in figure 2. This initial analysis suggests that there is a strong relationship between all the CDR layers when the data are aggregated over a long period of time. However, aggregating the data over a period of two months may remove differences which appear at different time scales. To investigate this further, we then aggregate the data at two different temporal levels: daily and hourly. For the daily aggregation, we effectively construct 62 snapshots for each CDR layer across the whole spatial grid; for the hourly aggregation, we have 1488 snapshots, corresponding to the number of hours in the dataset. We then want to investigate whether there are differences in the relationship between CDR layers at the daily and hourly level. Figure 3a,b depicts the relationship between the number of SMSs received and the Internet activity across all cells for different time periods. At the daily level (figure 3b), we again note a strong similarity between the CDR layers even on different days. However, when looking at hourly data on different days and different times of the day (figure 3a), we see the appearance of strong differences between the number of SMSs received and the Internet activity. Figure 3c,d shows how the correlation between the number of SMSs received and Internet activity evolves over time. We find again that, when considering hourly data (figure 3c), there is a strong variation in the strength of this relationship, with the Kendall's correlation values ranging from less than 0.5 to more than 0.8. When considering daily data (figure 3d), we observe a decrease in the strength of the relationship, as well as some variations, but it is important to note that the range of these variations

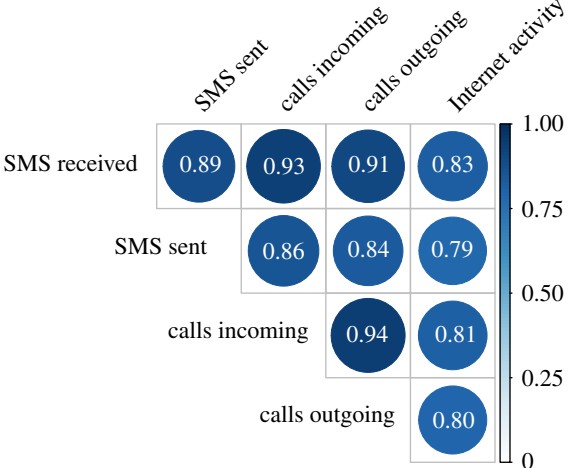

**Figure 2.** Similarity of aggregate CDRs. We aggregate all CDRs over the period of analysis to perform a preliminary investigation of the relationship between the different layers. Here, we depict Kendall's correlation coefficient. We find that, when aggregating temporally, all CDR layers have a strong positive correlation with each other. We also note that there is some variation in the strength of this relationship between different layers.

is relatively small, with the correlation coefficient values ranging from just below 0.80 to around 0.85. Similar results can be observed by fitting a linear regression model at each hour (day), and plotting the evolution of the regression coefficient over time (figure 3*e*,*f*). In the linear regression, we have considered the number of SMSs received as the independent variable, and Internet activity as dependent variable. It is important to comment on the variation in the regression coefficient since it gives us an indication of the ratio between number of SMSs received and Internet activity. This is an interesting result since it highlights how the relationship between CDR layers can vary drastically over time when considering hourly data. This highlights the crucial need for retraining different models when using data derived from different CDR layers, as well as the need for a general better understanding of the behavioural differences in how we use mobile phones. The results just presented for one combination of two CDR layers hold across all pairwise combinations, and the corresponding figures are presented in the electronic supplementary material.

Our analysis so far has investigated the relationship between different CDR layers by looking at individual time points, either hours or days, and assessing how the relationship changed between different time points. We now take a different approach and construct one time series for each cell and each CDR layer for the whole period of analysis. For instance, for a given cell, we have five different time series corresponding to the five different CDR layers we consider. For consistency with the previous approach, we consider the time series both at hourly aggregation and daily aggregation. Figure 4 depicts the hourly time series of all five CDR layers for one specific cell in our spatial grid. It is important to emphasize that this is only one particular cell, and, given the spatial heterogeneity of the different CDR layers, this is not representative of every cell. However, visual inspection of this specific case allows us to discuss several interesting features. Broadly speaking, all time series have similar patterns, with strong regularities due to the regular behaviour of our daily and weekly life patterns. However, we also observe some notable differences, such as the clear difference in the behaviour during the Christmas period of the different CDR layers. For instance, the Internet activity presents a sharp decrease in this period, much more remarked than other CDR layers. To quantify these differences between time series, we carry out a further analysis. For each cell, we calculate the pairwise distance between each time series using the distance measures specified in the *Methods* section above. We repeat this for each cell in the spatial grid, thus resulting in 10 different grids of 10 000 cells corresponding to the pairwise distances of the five CDR layers in each cell. Figure 5 (top row) depicts the relationship between two of these grids. Visual inspection clearly shows a positive correlation, suggesting that cells which have a large distance between two CDR layers have a large distance between other CDR layers too. This provides evidence that, for these cells, all CDR layers are different from one another and that the dissimilarity between CDR layers is not limited to only two layers. However, it is also worth noting that many cells have low distance between multiple layers. This is also highlighted in figure 5 (bottom row) where the pairwise correlation is displayed, confirming our previous intuition that there is a positive correlation between the various distance measures. Analogous results hold when we aggregate the time series at the daily level, and the corresponding figures are presented in the electronic supplementary material.

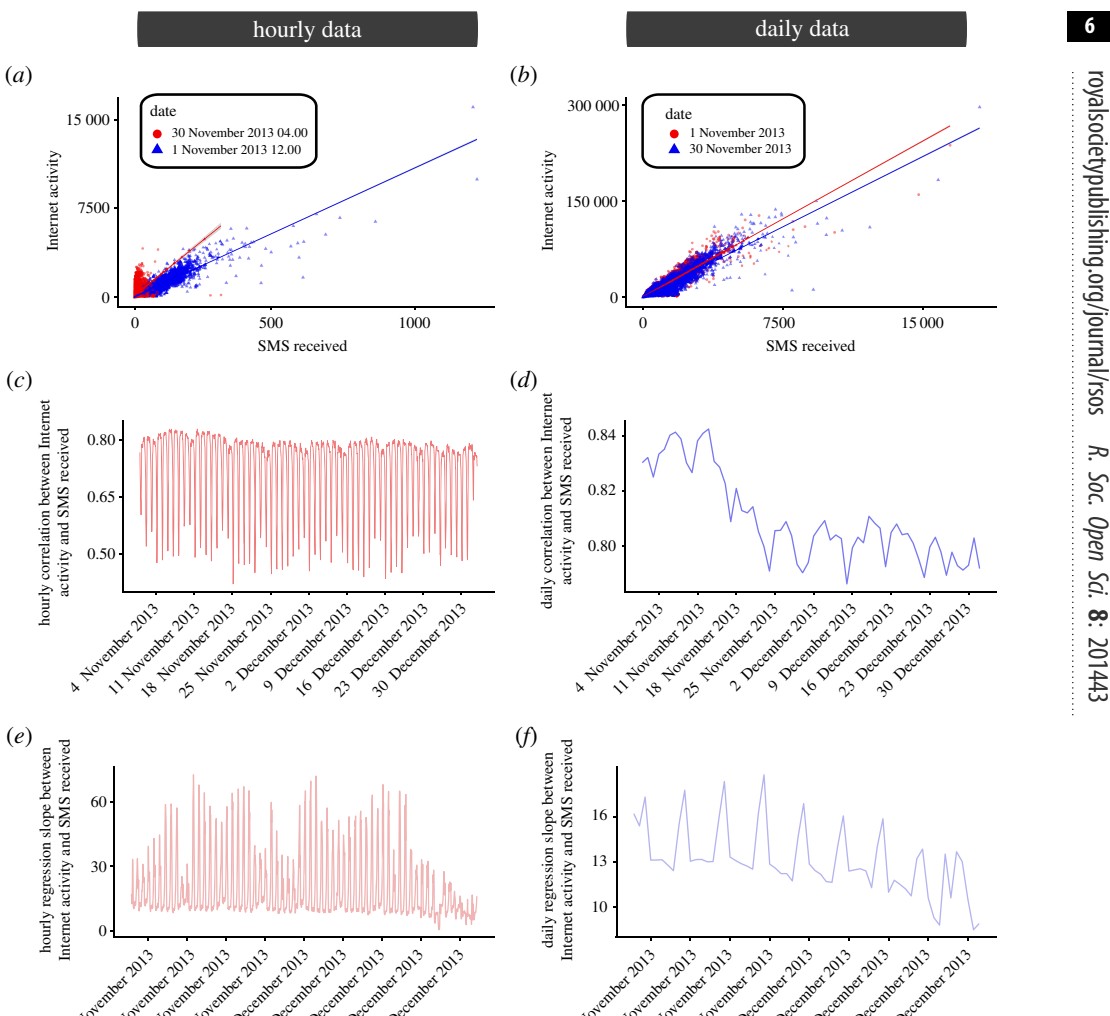

**Figure 3.** Evolution of the relationship between CDRs. We aim to investigate whether the relationship between different CDRs is consistent throughout the period and area under analysis. (*a,b*) We depict here the relationship between SMSs received and Internet activity in each cell for the different time periods specified in the plots legend. When analysing hourly data (*a*), visual inspection indicates that the relationship differs between daytime hours and night-time hours; when looking at daily data (*b*) on the other hand, the relationship between different days seems to be more consistent. (*c,d*) For each hour (*c*), or day (*d*), we calculate Kendall's correlation coefficient between Internet activity and SMSs received, and we plot it here for the whole period of analysis. All correlation values are significant at the 5% level, and all *p*-values have been corrected using *false discovery rate* correction to account for multiple hypothesis testing. We note that the evolution of the hourly correlation (*c*) presents strong and clear patterns, with the strength of the correlation oscillating from less than 0.5 to more than 0.8; while the daily correlation (*d*) also presents differences in the period of analysis, it is important to note that the variations are in a relatively small range of values. (*e,f*) Similar to plots (*c,d*), we depict here the evolution of the regression coefficient between SMSs received (independent variable) and Internet activity (dependent variable) in each cell. Again, we note clear patterns both in the hourly (*e*) as well as in the daily (*f*) data.

Finally, so far we have analysed the temporal differences between the time series, but we have not considered the spatial distribution of these differences. To study this, we rely on *Moran's I* to measure the spatial correlation, if any, present in these differences [39]. For each of the 10 grids just discussed, each representing the distance between two different CDR layers, we run Moran's test for spatial autocorrelation, and we do this for each of the three distance measures used above. We assume two cells to be connected if they share a common boundary, i.e. they are neighbours. We find a significant positive spatial correlation in all 10 grids regardless of the distance measure used, both at the hourly (all $I > 0.335$; all $p < 0.001$, *fdr* corrected; $N = 10\,000$) and the daily aggregation level (all $I > 0.367$; all $p < 0.001$, *fdr* corrected; $N = 10\,000$). This analysis shows the existence of global spatial autocorrelation, but provides no information on whether it is homogeneous across the area under analysis. To further investigate this, we calculate the local spatial correlation by using the local version of Moran's I coefficient, assuming again that two cells are connected if they share a common boundary. For both

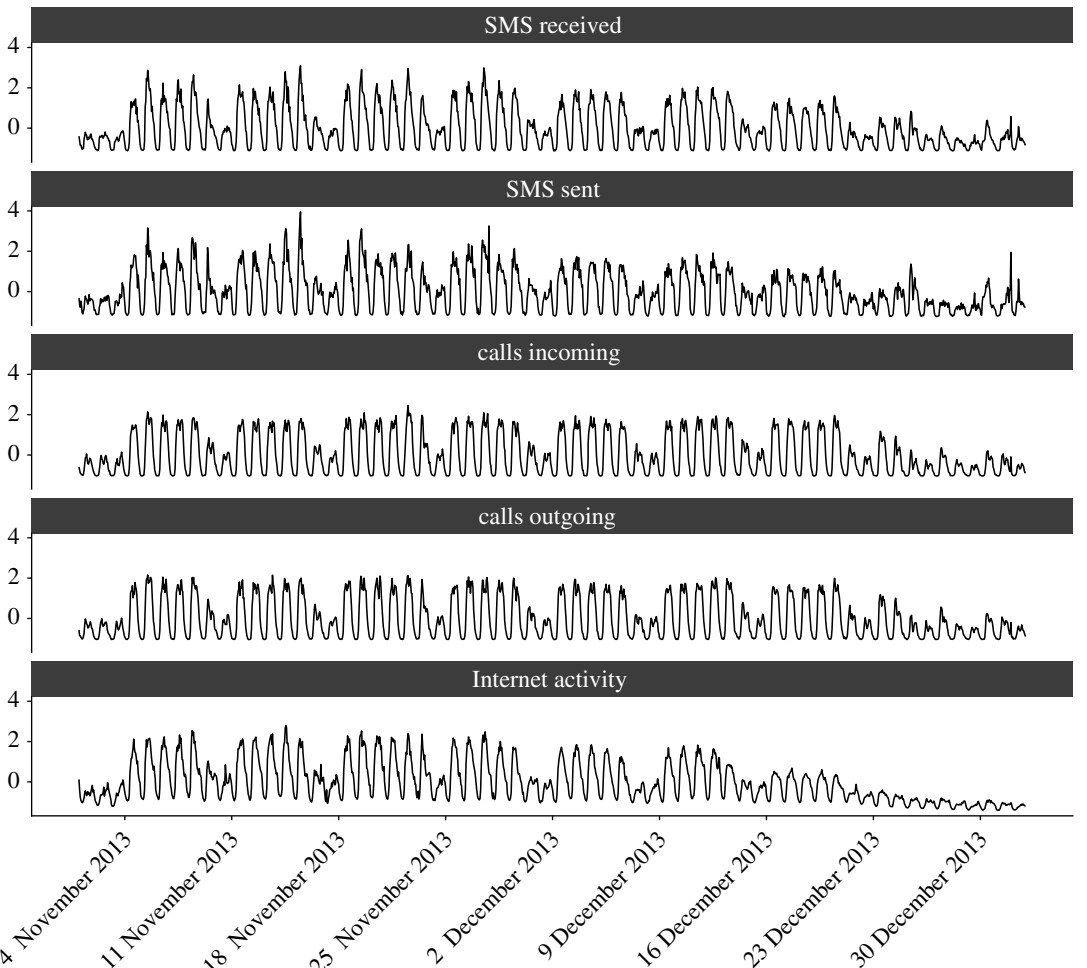

**Figure 4.** Differences in the temporal evolution of CDRs. Here, we focus on a specific cell in the area under analysis, and depict the hourly evolution of the different CDRs in our dataset. In our analysis, we aim to quantify the differences and the distance between these time series in each cell, and investigate whether these differences are consistent across time. In this example, we note that the broad patterns of the time series are similar, but differences can be spotted. For instance, the Internet activity has a distinct pattern in the last 10 days of analysis, with the values exhibiting a sharp decrease compared to the rest. Crucially, this decrease is more remarked than the decrease observed in the other CDRs. It is important to highlight that the data used for this figure refers to only one specific cell in the dataset, which is not necessarily representative of the whole grid, given spatial heterogeneities.

hourly and daily aggregation, and for each distance measure, we calculate the local spatial correlation between the distance of two CDR layers. Figure 6 depicts the results for the distance between incoming SMS and Internet for all distance measures and for both hourly (*a–c*) and daily (*d–f*) temporal aggregation. This shows interesting results. In the case of Kendall's correlation as a distance measure, the spatial correlation is distributed in clusters across the whole area of analysis. However, for the other two distance measures, we find that the spatial correlation is strongly clustered in the city centre. Note that, for visualization purposes, values of the local spatial correlation which are not statistically significant (after *fdr* correction) have been assigned a value of 0, and the maps are plotted on a logarithmic scale. Results for the other CDR layers are included in the electronic supplementary material.

# 5. Discussion and conclusion

Our analysis has highlighted some of the important features and limitations that need to be considered when analysing CDR data. First, generally speaking we have shown that different CDR layers are broadly similar if they are aggregated over a sufficiently long period of time (figure 2). However, we have also shown that this similarity varies rather drastically if the same data is aggregated over shorter periods of time (figure 3). This indicates that small differences, which are present at small temporal granularity, are 'washed away' when aggregating over long periods of time. This is an important

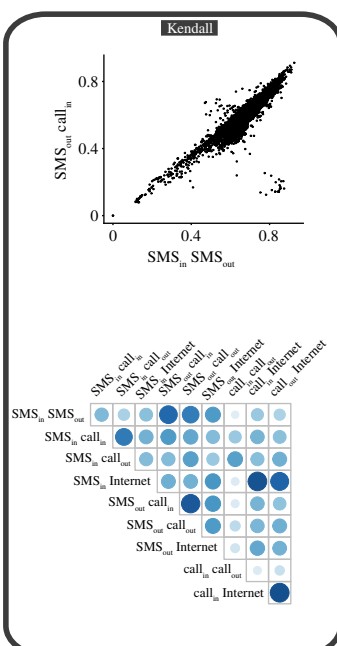
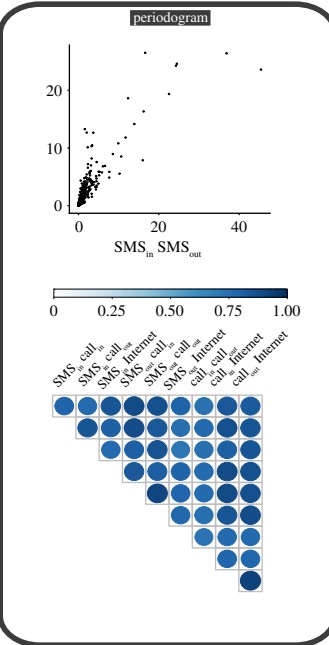
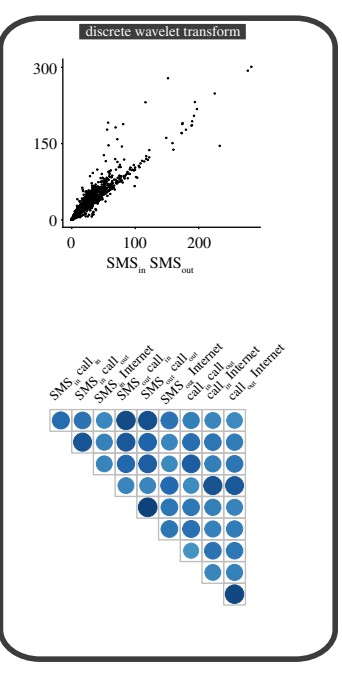

**Figure 5.** Differences in CDR layers. For each cell, we construct five time series, corresponding to the different CDR layers, at hourly granularity, and we calculate the pairwise distances between them using three different distance measures: Kendall's correlation, distance between the time-series periodogram coefficients, and the discrete wavelet transform. (Top row) We depict here the relationship between the pairwise distance between the number of sent and received SMSs ($x$-axis), and the pairwise distance between sent SMSs and incoming calls ($y$-axis). Each point in the scatter plot corresponds to a different cell in the spatial grid. Visual inspection clearly indicates that cells with a large distance between two CDR layers also tend to have large distance between two other CDR layers, indicating an overall dissimilarity between all CDR layers for some cells. (Bottom row) We further investigate whether the visual findings are consistent across all cells and time series. The correlation matrices report the values of the correlation between the pairwise distances across all cells. We find that cells for which there is a larger distance between two time series correspond to cells with a larger distance across all time series. This provides further evidence supporting our previous hypothesis. It is also important to note that, while for some cells all CDR layers may be dissimilar, for many other cells this does not happen and different CDR layers are close to one another.

consideration, since analyses performed at different levels of temporal aggregation may or may not be affected by this. Our results hold across all CDR layers considered in the analysis, including CDRs generated from Internet usage. The analysis of Internet activity in particular extends previous results which had already highlighted the existence of differences between mobile phone interactions data [30–33].

We have also seen that there exist cells for which the different CDR layers are consistently dissimilar from one another, even though the majority of cells tend to exhibit similar behaviour across CDR layers. This is important because, compared to existing studies, we have not used individual-level data. The existence of these cells needs to be carefully assessed when trying to generalize results obtained in one area to a different one. Again, while this may not be of concern in many situations, there undoubtedly are scenarios in which this limitation may be critical. Additionally, the existence of spatial clustering also poses some challenges since even spatial aggregation of the data may still suffer from similar issues. The existence of spatial clustering is in line with previous studies which have shown that mobile phone interactions tend be more likely to happen between individuals who are in spatial proximity [31,40]. Further work should investigate whether spatial clusters are linked with the urban features of the areas where the clusters appear. This may provide interesting information on the connections between mobile phone interactions and urban environments.

More broadly, the limitations highlighted here are relevant for more general applications which rely on new forms of data. Since these datasets are inherently related to our collective and individual behaviour, any changes in the way we behave is bound to affect the data. Crucially, these new forms of data often rely on specific platforms, such as social media, which may only be popular for short periods of time, or for which the user sample may vary over the years. A key challenge then is to understand how these different forms of data can be integrated or even replaced should any of them not be available any more. This is particularly important for data sources that are privately owned, which may be made publicly or privately accessible at discretion of the company with practically no notice.

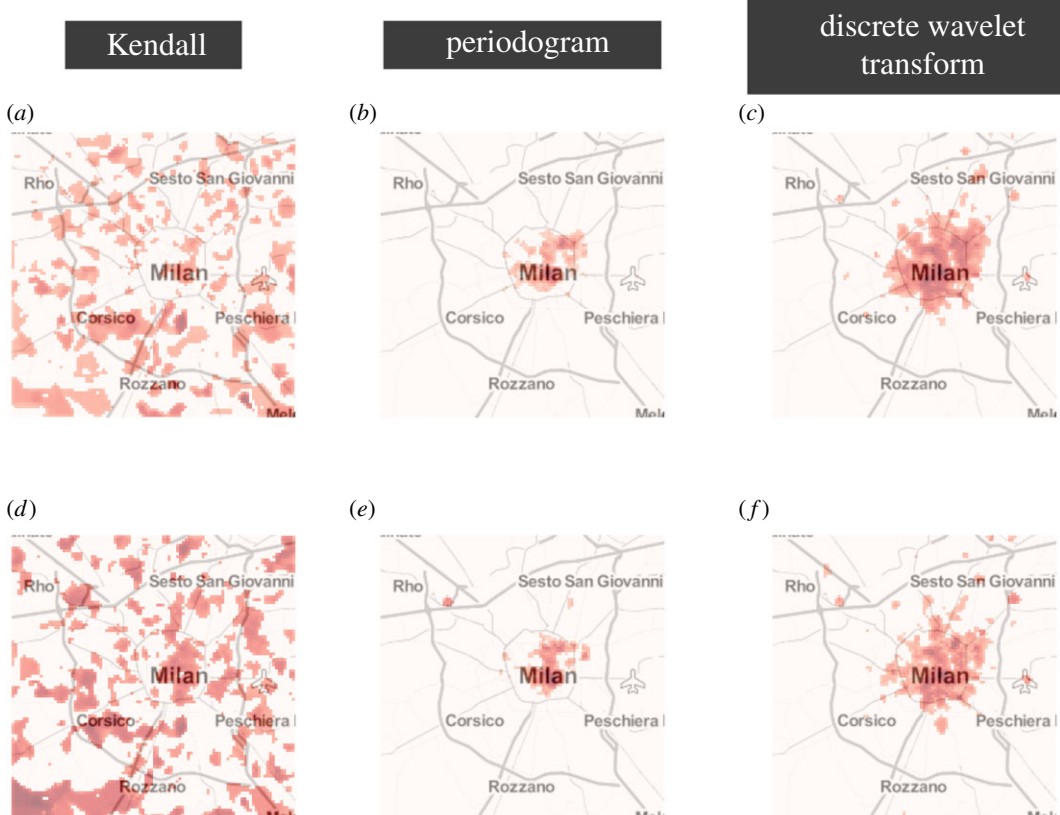

**Figure 6.** Local spatial correlation. For each cell, we construct five time series, corresponding to the different CDR layers, at hourly and daily granularity, and we calculate the pairwise distances between them using the three different distance measures discussed above. This gives rise to 10 spatial grids each corresponding to the pairwise distance at either hourly or daily aggregation. For each of these grids, we then calculate the local Moran's I coefficient by considering the neighbouring cells of each cell. We depict here three of the resulting maps depicting the local Moran's I coefficients for the spatial autocorrelation in the three distance measures between the incoming SMS layer and the Internet layer both at the hourly aggregation (*a–c*) and the daily aggregation (*d–f*). For visualization purposes, the maps are presented on a logarithmic scale and local Moran's I coefficients which are not statistically significant have been set to have value 0 in this scale (all *p*-values have been adjusted using *fdr* correction). Darker shades of red correspond to larger values of the local Moran's I coefficient.

It is also important to acknowledge the limitations of our study. First of all, the dataset used here is limited to a two-month time period, one city and one mobile phone provider. In fact, future work should explore the existence of even more remarked differences between CDRs of different mobile phone providers, since pricing plans of different providers may encourage different behaviour in their users. Additionally, some of the observed differences in CDR data may also arise naturally due to the different nature of the various CDR layers analysed here. For instance, SMS data are inherently more sparse compared to Internet data, which is generated more continuously, and passively, by mobile phones. It is also important to stress that different populations, as well as different cultures, may have different behaviours that we could not observe in our data. Second, our analysis has largely focused on the differences in CDR data at various levels of temporal aggregation. Further work could be done to analyse how the spatial aggregation may affect the data. Future work should also look more closely at differences that arise when considering weekdays and weekends separately, as these will undoubtedly give rise to interesting behavioural differences that will be reflected in CDR data. Finally, more work needs to be done to understand the behavioural features which give rise to differences in the use of phones and different communications channel.

While the limitations highlighted in this study are important to consider when working with this kind of data, we believe that the use of mobile phone data offers unique opportunities in the study of our behaviour and society. Going forward, we envisage more and more researchers relying on mobile phone data to study human behaviour both in developed and developing countries. We anticipate that the considerations highlighted in our study will be of use in assessing the limitations and generalizability of such studies. Finally, we expect similar considerations to be valid also for different

social media platforms. Further work should explore the similarity and differences between data derived from different social media sites.

Data accessibility. All data used in the analysis is publicly available in this publication [35].
Competing interests. We declare we have no competing interests.
Funding. No funding has been received for this article.

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
