## [Peer Review File · Royal Society Open Science]

Review History

RSOS-201443.R0 (Original submission)

Review form: Reviewer 1

Is the manuscript scientifically sound in its present form?

Yes

Are the interpretations and conclusions justified by the results?

Yes

Is the language acceptable?

Yes

Do you have any ethical concerns with this paper?

No

Have you any concerns about statistical analyses in this paper?

No

Recommendation?

Major revision is needed (please make suggestions in comments)

Comments to the Author(s)

Please see attached file (Appendix A).

Review form: Reviewer 2 (Matteo Zignani)**Is the manuscript scientifically sound in its present form?**

Yes

Are the interpretations and conclusions justified by the results?

Yes

Is the language acceptable?

Yes

Do you have any ethical concerns with this paper?

No

Have you any concerns about statistical analyses in this paper?

No

Recommendation?

Accept with minor revision (please list in comments)

Comments to the Author(s)

The paper addresses the issue of the spatio-temporal heterogeneity of mobile phone data, specifically Call Detail Records (CDRs), and the effect of the temporal and spatial aggregations when comparing the different communication/usage channels, i.e. text messages, calls and internet activity. The study is interesting and maybe a good reference point for researchers interested in computational social science, since CDRs are becoming one of the fundamental data source for the comprehension of many social issues on large scale populations. The main merit of the work is that it systematically and quantitatively address a problem which many researchers dealing with mobile phone data have tackled. CDR data are extremely heterogenous on different dimensions; not only in their spatio-temporal aspects, but also in theirs social aspects, i.e. how people interact through different communication channels. However, it does not represent a novelty, since many studies in the computer and mobile network community dealt with these heterogeneity - see for instance "Naboulsi, D., Fiore, M., Ribot, S., & Stanica, R. (2015). Large-scale mobile traffic analysis: a survey. *IEEE Communications Surveys & Tutorials*, 18(1), 124-161". In summary, the work may express its best potentiality for the computational social science audience.

As for the methodology, it sounds and rely on a public available dataset - even if it is bit outdate - , making easy to reproduce and verify the results presented in the paper.

For these reasons I would suggest a minor revision referencing to previous works which have addressed and highlighted the differences in CDRs when dealing with the different layers of interaction.

Here a few suggestions:

- Naboulsi, D., Fiore, M., Ribot, S., & Stanica, R. (2015). Large-scale mobile traffic analysis: a survey. *IEEE Communications Surveys & Tutorials*, 18(1), 124-161

- Heydari, S., Roberts, S. G., Dunbar, R. I., & Saramäki, J. (2018). Multichannel social signatures and persistent features of ego networks. *Applied network science*, 3(1), 8.
- A. A. Nanavati, R. Singh, D. Chakraborty, K. Dasgupta, S. Mukherjea, G. Das, S. Gurumurthy, and A. Joshi, "Analyzing the structure and evolution of massive telecom graphs," *Knowledge and Data Engineering, IEEE Transactions on*, vol. 20, no. 5, pp. 703-718, 2008
- Zignani, M., Quadri, C., Gaito, S., & Rossi, G. P. (2015). Calling, texting, and moving: multidimensional interactions of mobile phone users. *Computational Social Networks*, 2(1), 13.

Decision letter (RSOS-201443.R0)

Dear Dr Botta

The Editors assigned to your paper RSOS-201443 "Quantifying the differences in Call Detail Records." have now received comments from reviewers and would like you to revise the paper in accordance with the reviewer comments and any comments from the Editors. Please note this decision does not guarantee eventual acceptance.

Please submit your revised manuscript and required files (see below) no later than 21 days from today's (ie 30-Nov-2020) date. Note: the ScholarOne system will 'lock' if submission of the revision is attempted 21 or more days after the deadline. If you do not think you will be able to meet this deadline please contact the editorial office immediately.

on behalf of Dr Mirco Musolesi (Associate Editor) and Marta Kwiatkowska (Subject Editor)
 openscience@royalsociety.org

Associate Editor Comments to Author (Dr Mirco Musolesi):

Associate Editor: 1

Comments to the Author:

The reviewers raised some valid concerns that should be addressed in a major revision of the manuscript. In particular, there are some comments about aspects of the analysis and the presentation of the results that should be taken into consideration very carefully.

Reviewer comments to Author:

Reviewer: 1

Comments to the Author(s)

Please see attached file.

Reviewer: 2

Comments to the Author(s)

The paper addresses the issue of the spatio-temporal heterogeneity of mobile phone data, specifically Call Detail Records (CDRs), and the effect of the temporal and spatial aggregations when comparing the different communication/usage channels, i.e. text messages, calls and internet activity. The study is interesting and maybe a good reference point for researchers interested in computational social science, since CDRs are becoming one of the fundamental data source for the comprehension of many social issues on large scale populations. The main merit of the work is that it systematically and quantitatively address a problem which many researchers dealing with mobile phone data have tackled. CDR data are extremely heterogenous on different dimensions; not only in their spatio-temporal aspects, but also in their social aspects, i.e. how people interact through different communication channels. However, it does not represent a novelty, since many studies in the computer and mobile network community dealt with these heterogeneity - see for instance "Naboulsi, D., Fiore, M., Ribot, S., & Stanica, R. (2015). Large-scale mobile traffic analysis: a survey. *IEEE Communications Surveys & Tutorials*, 18(1), 124-161". In summary, the work may express its best potentiality for the computational social science audience.

As for the methodology, it sounds and rely on a public available dataset - even if it is bit outdate - , making easy to reproduce and verify the results presented in the paper.

For these reasons I would suggest a minor revision referencing to previous works which have addressed and highlighted the differences in CDRs when dealing with the different layers of interaction.

Here a few suggestions:

- Naboulsi, D., Fiore, M., Ribot, S., & Stanica, R. (2015). Large-scale mobile traffic analysis: a survey. *IEEE Communications Surveys & Tutorials*, 18(1), 124-161
- Heydari, S., Roberts, S. G., Dunbar, R. I., & Saramäki, J. (2018). Multichannel social signatures and persistent features of ego networks. *Applied network science*, 3(1), 8.
- A. A. Nanavati, R. Singh, D. Chakraborty, K. Dasgupta, S. Mukherjea, G. Das, S. Gurumurthy, and A. Joshi, "Analyzing the structure and evolution of massive telecom graphs," *Knowledge and Data Engineering*, *IEEE Transactions on*, vol. 20, no. 5, pp. 703-718, 2008
- Zignani, M., Quadri, C., Gaito, S., & Rossi, G. P. (2015). Calling, texting, and moving: multidimensional interactions of mobile phone users. *Computational Social Networks*, 2(1), 13.

===PREPARING YOUR MANUSCRIPT===

===PREPARING YOUR REVISION IN SCHOLARONE===

- Any electronic supplementary material (ESM).
- If you are requesting a discretionary waiver for the article processing charge, the waiver form must be included at this step.
- If you are providing image files for potential cover images, please upload these at this step, and inform the editorial office you have done so. You must hold the copyright to any image provided.
- A copy of your point-by-point response to referees and Editors. This will expedite the preparation of your proof.

- Ensure that your data access statement meets the requirements at <https://royalsociety.org/journals/authors/author-guidelines/#data>. You should ensure that you cite the dataset in your reference list. If you have deposited data etc in the Dryad repository, please include both the 'For publication' link and 'For review' link at this stage.
- If you are requesting an article processing charge waiver, you must select the relevant waiver option (if requesting a discretionary waiver, the form should have been uploaded at Step 3 'File upload' above).
- If you have uploaded ESM files, please ensure you follow the guidance at <https://royalsociety.org/journals/authors/author-guidelines/#supplementary-material> to include a suitable title and informative caption. An example of appropriate titling and captioning may be found at https://figshare.com/articles/Table_S2_from_Is_there_a_trade-off_between_peak_performance_and_performance_breadth_across_temperatures_for_aerobic_scooping_in_teleost_fishes_/3843624.

Author's Response to Decision Letter for (RSOS-201443.R0)

See Appendix B.

RSOS-201443.R1 (Revision)

Review form: Reviewer 1

Is the manuscript scientifically sound in its present form?

Yes

Are the interpretations and conclusions justified by the results?

Yes

Is the language acceptable?

Yes

Do you have any ethical concerns with this paper?

No

Have you any concerns about statistical analyses in this paper?

No

Recommendation?

Accept with minor revision (please list in comments)

Comments to the Author(s)

Please see the attached report (Appendix C).

Decision letter (RSOS-201443.R1)

Dear Dr Botta

On behalf of the Editors, we are pleased to inform you that your Manuscript RSOS-201443.R1 "Quantifying the differences in Call Detail Records." has been accepted for publication in Royal Society Open Science subject to minor revision in accordance with the referees' reports. Please find the referees' comments along with any feedback from the Editors below my signature.

Please submit your revised manuscript and required files (see below) no later than 7 days from today's (ie 15-Mar-2021) date. Note: the ScholarOne system will 'lock' if submission of the revision is attempted 7 or more days after the deadline. If you do not think you will be able to meet this deadline please contact the editorial office immediately.

Kind regards,

Royal Society Open Science Editorial Office
Royal Society Open Science
openscience@royalsociety.org

on behalf of Dr Mirco Musolesi (Associate Editor) and Marta Kwiatkowska (Subject Editor)
openscience@royalsociety.org

Associate Editor Comments to Author (Dr Mirco Musolesi):

Associate Editor: 1

Comments to the Author:

The author addressed all the major remaining issues and, for this reason, I would recommend the paper for publication. Having said that, there are still some required changes that have been suggested by one of the reviewers - these should be addressed before publication.

Associate Editor: 2

Comments to the Author:

(There are no comments.)

Reviewer comments to Author:

Reviewer: 1

Comments to the Author(s)

Please see the attached report.

===PREPARING YOUR MANUSCRIPT===

===PREPARING YOUR REVISION IN SCHOLARONE===

To revise your manuscript, log into <https://mc.manuscriptcentral.com/rsos> and enter your Author Centre - this may be accessed by clicking on "Author" in the dark toolbar at the top of the

page (just below the journal name). You will find your manuscript listed under "Manuscripts with Decisions". Under "Actions", click on "Create a Revision".

<https://royalsociety.org/journals/authors/author-guidelines/#supplementary-material> to include a suitable title and informative caption. An example of appropriate titling and captioning may be found at https://figshare.com/articles/Table_S2_from_Is_there_a_trade-off_between_peak_performance_and_performance_breadth_across_temperatures_for_aerobic_sc_ope_in_teleost_fishes_/3843624.

Author's Response to Decision Letter for (RSOS-201443.R1)

See Appendix D.

RSOS-201443.R2 (Revision)

Review form: Reviewer 1

Is the manuscript scientifically sound in its present form?

Yes

Are the interpretations and conclusions justified by the results?

Yes

Is the language acceptable?

Yes

Do you have any ethical concerns with this paper?

No

Have you any concerns about statistical analyses in this paper?

No

Recommendation?

Accept as is

Comments to the Author(s)

Thank you for addressing my comments and congrats for your paper!

Decision letter (RSOS-201443.R2)

Dear Dr Botta,

It is a pleasure to accept your manuscript entitled "Quantifying the differences in Call Detail Records." in its current form for publication in Royal Society Open Science. The comments of the reviewer(s) who reviewed your manuscript are included at the foot of this letter.

You can expect to receive a proof of your article in the near future. Please contact the editorial office (openscience@royalsociety.org) and the production office (openscience_proofs@royalsociety.org) to let us know if you are likely to be away from e-mail contact – if you are going to be away, please nominate a co-author (if available) to manage the proofing process, and ensure they are copied into your email to the journal.

on behalf of Dr Mirco Musolesi (Associate Editor) and Marta Kwiatkowska (Subject Editor)
openscience@royalsociety.org

Associate Editor Comments to Author (Dr Mirco Musolesi):

Comments to the Author:

The author addresses all the remaining points - I would recommend the acceptance of this paper without any further modifications.

Reviewer comments to Author:

Reviewer: 1

Comments to the Author(s)

Thank you for addressing my comments and congrats for your paper!

Appendix A

Quantifying the differences in Call Detail Records

Reviewer report

This paper aims to compare the differences between the various CDR layers focusing mostly on the temporal dimension. The take home message is that the more granular the CDR data is, the more substantially the different CDR layers are. In general, the results are well supported by the analysis. I have some specific comments below, but my main concerns has to do with the state of the art. How new are the below findings? Have other researchers illustrated similar differences? Didn't we know already that granularity – both spatial and temporal – affects the observed patterns and that the different CDR layers represent different types of behaviour – e.g. SMS (which is not really relevant anymore at least for the more developed countries) vs. calls vs. internet traffic.

Please see below more detailed comments.

p. 2: How about population biases affecting the observed behavioural changes?

I have a preference of using the word internet with a lower case i. I do understand that different disciplines have different traditions and views on this, so I am just flagging this to the authors and leave the choice on them.

The quality of Figure 1 is too low for me to be able to assess it.

Daily data: do the authors differentiate between working and non-working days? This is very common in the relevant literature.

Technically speaking, what are the qualitative differences between the different CDR layers? For instance, SMS received are only registered when an SMS is received. But how does internet activity work? Does it refer to internet activity over a browser or does it include internet activity generated from apps? If it is the latter, I would expect and almost 'stream' type of data contrary to the more sparse observations based on SMS. Can these technical characteristics explain some of the differences between the CDR layers that you indicate in the paper, eg. in Figure 4?

Figure 3, C and D: are correlation coefficients between internet activity and SMS received? Please clarify in the figure description.

Can you please provide an equation of the linear model or just clarify left/right hand side variables? Why did you choose to estimate a linear model instead of presenting the correlation coefficients?

How many cells are included in the study area? Some descriptive statistics would be useful.

Figure 4: one might speculate and say that this is a cell from an office area. Given the spatial heterogeneity of the mobile phone usage, how useful is to present the 5 layers of mobile traffic for just one cell?

In our analysis, we aim to quantify the differences and the distance between these time series in each cell, and investigate whether these differences are consistent across space and time.

As per the limitations paragraph, the paper does not equally focus on space and time.

Figure 5: I am afraid this is not entirely clear to me. The authors calculate the pairwise differences between the time series depicting the 5 CDR layers. For instance, the pairwise difference between in and out SMS. What is the scope of plotting scatterplots between two differences? To be more specific, what can we learn by comparing the differences between the distance between, let's say SMSout and Callin vs. SMSin and SMSout?

Is there a scope of also implementing LISA (Local Indicator of Spatial Autocorrelation) in order provide more spatial details about the SI and reveal potentially interesting clusters? So, is this autocorrelation homogeneous across the study area cells or not?

Main finding:

that small differences, which are present at small temporal granularity, are "washed away" when aggregating over long periods of time.

I agree with this and it is supported by the analysis, but I wonder what is already known in the literature on this topic? I suspect that this is common knowledge both in the time series literature and in empirical studies using data from mobile operators for urban analytics. Please clarify if and how your findings are new considering the previous literature?

We have also seen that there exist cells for which the different CDR layers are consistently dissimilar from one another, even though the majority of cells tend to exhibit similar behaviour across CDR layers.

Again, is this a new finding? What do we already know regarding this? There is so much spatial analysis on CDR. How are these findings new?

For instance, this may be the case when trying to generalise a model trained on SMS data alone to one which also includes calls data.

Fair point, but do you have any examples of such issues? It is difficult to imagine such a fallacy.

Appendix B

Federico Botta
Lecturer in Data Science

College of Engineering, Mathematics and
Physical Sciences
Harrison Building
Streatham Campus
University of Exeter
North Park Road
UK, EX4 4QF

+44 (0)1392 722220
f.botta@exeter.ac.uk

04 January 2021

Revision of "Quantifying the differences in *Call Detail Records*"

Dear Dr Mirco Musolesi (Associate Editor) and Marta Kwiatkowska (Subject Editor),

We were extremely pleased to receive your decision letter on the 30th November 2020 with the positive feedback from the two reviewers. We were glad to hear that the reviewers agreed that the results were interesting, supported by the analysis and well written.

The reviewers' feedback was very useful indeed and has allowed us to further improve the quality of our manuscript. We include below here the comments of the reviewers, as well as our specific responses to each of their points. Additionally, we also include a PDF copy of our manuscript which highlights the changes we did to our manuscript as a result of the feedback from the reviewers.

We greatly appreciate the time taken by the reviewers to provide helpful and constructive feedback, which we are confident we have addressed in our revised manuscript. We hope that our manuscript is now ready to be published in *Royal Society Open Science*.

Thank you once again for your kind help and consideration in this matter.

Sincerely,

Federico Botta

Reviewer #1:

1. This paper aims to compare the differences between the various CDR layers focusing mostly on the temporal dimension. The take home message is that the more granular the CDR data is, the more substantially the different CDR layers are. In general, the results are well supported by the analysis.

We thank the reviewer for their general feedback on the manuscript as well as for agreeing that our results are well supported by the analysis.

2. I have some specific comments below, but my main concerns has to do with the state of the art. How new are the below findings? Have other researchers illustrated similar differences? Didn't we know already that granularity – both spatial and temporal – affects the observed patterns and that the different CDR layers represent different types of behaviour – e.g. SMS (which is not really relevant anymore at least for the more developed countries) vs. calls vs. internet traffic.

We thank the reviewer for their comment on this important topic. We have expanded our discussion across the whole manuscript, and in particular in the *Introduction* and *Discussion and Conclusion* sections, to clarify the relationship between our results and the existing literature.

3. How about population biases affecting the observed behavioural changes?

We agree with the reviewer that population biases may indeed be important in observing behavioural changes and differences. We have expanded our introduction, as well as the discussion on limitations, to provide a more direct reference to the possibility of population biases in our data.

4. I have a preference of using the word internet with a lower case i. I do understand that different disciplines have different traditions and views on this, so I am just flagging this to the authors and leave the choice on them.

We thank the reviewer for bringing this to our attention. For consistency with other papers in the computational social science area, we have decided to keep the capital i. However, we appreciate the comment from the reviewer on this.

5. The quality of Figure 1 is too low for me to be able to assess it.

We thank the reviewer for pointing out about the low quality of Figure 1. We have recreated the figure and have removed the grids from the maps with the CDR data, in order to enhance clarity and improve the quality of the figure.

6. Daily data: do the authors differentiate between working and non-working days? This is very common in the relevant literature.

We agree with the reviewer that this is a potentially very interesting direction to explore, which, however, we feel goes beyond the scope of the current study. We have clarified in the Data section that we do not differentiate between weekdays and weekends, and we have also highlighted this in the discussion on the limitations and areas of future work in the Conclusion section.

7. Technically speaking, what are the qualitative differences between the different CDR layers? For instance, SMS received are only registered when an SMS is received. But how does internet activity work? Does it refer to internet activity over a browser or does it include internet activity generated from apps? If it is the latter, I would expect and almost 'stream' type of data contrary to the more sparse observations based on SMS.

We thank the reviewer for pointing out that we could have given a better qualitative overview of how the different CDR layers data is generated. We have expanded the Data section to include a better description of the data, and in particular of how the Internet CDR layer has been calculated by the mobile phone provider.

8. Can these technical characteristics explain some of the differences between the CDR layers that you indicate in the paper, eg. in Figure 4?

This is a very interesting point, which we hadn't discussed enough in our original submission. It may indeed be the case that some differences arise due to the way the data is generated, and this should be further explored in the future. We have expanded our discussion on this in the limitations in the Conclusion section.

9. Figure 3, C and D: are correlation coefficients between internet activity and SMS received? Please clarify in the figure description.

We thank the reviewer for pointing out that the figure description for Figure 3 C-D could have been clearer. We have expanded the description to improve clarity and provide all the relevant information needed for the reader to understand the figure. We have also improved the presentation of this in the main text of the manuscript.

10. Can you please provide an equation of the linear model or just clarify left/right hand side variables? Why did you choose to estimate a linear model instead of presenting the correlation coefficients?

We thank the reviewer for highlighting that the presentation of the linear models could be improved. We have clarified which variables enter the model as independent or dependent both in the description of Figure 3, in the Results section and in the Supplementary Material.

11. How many cells are included in the study area? Some descriptive statistics would be useful.

We thank the reviewer for highlighting that our presentation of the study area and how it is divided could be improved. To address their comment, we have expanded the Data section to better present the spatial grid used in our analysis and we have explicitly included the number of cells in the area.

12. Figure 4: one might speculate and say that this is a cell from an office area. Given the spatial heterogeneity of the mobile phone usage, how useful is to present the 5 layers of mobile traffic for just one cell?

We thank the reviewer for raising this issue, which wasn't clearly explained. We agree with the reviewer's comment that the figure is not necessarily representative of the full dataset, given the spatial heterogeneity. However, we believe that it allows to discuss some interesting points in our analysis. We have clarified this in the Results section and in the caption of Figure 4, explaining that the cell used to generate the figure is not necessarily representative of the full dataset and that it is used only as an example for our discussion. We think that this should avoid any confusion to the reader.

13. "In our analysis, we aim to quantify the differences and the distance between these time series in each cell, and investigate whether these differences are consistent across space and time."

As per the limitations paragraph, the paper does not equally focus on space and time.

We have removed the statement about differences in space from the caption of Figure 4, as we agree that it could have been confusing and unclear.

14. Figure 5: I am afraid this is not entirely clear to me. The authors calculate the pairwise differences between the time series depicting the 5 CDR layers. For instance, the pairwise difference between in and out SMS. What is the scope of plotting scatterplots between two differences? To be more specific, what can we learn by comparing the differences between the distance between, let's say SMSout and Callin vs. SMSin and SMSout?

We thank the reviewer for their comment on this result. We have edited the discussion on this result in the Results section, as well as in the caption to figure 5, in order to improve clarity and better present the findings of our analysis. The key message of this part of the analysis is that there is a positive correlation between the distances between different CDR layers, indicating that for some cells the differences are not limited to one or two specific CDR layers. We think that the revised explanation of this result is now clearer and provides a better insight into our analysis.

15. Is there a scope of also implementing LISA (Local Indicator of Spatial Autocorrelation) in order provide more spatial details about the SI and reveal potentially interesting clusters? So, is this autocorrelation homogeneous across the study area cells or not?

This is indeed a good suggestion since looking at the distribution of the spatial correlation provides additional information about spatial differences between CDR layers. We have calculated a local version of Moran's I coefficient, and included new figures, both in the main manuscript (Figure 6) and in the supplementary material (Figures S11-S19), to present the results. We have also included a discussion of these new results in the Results section.

16. Main finding:

“that small differences, which are present at small temporal granularity, are “washed away” when aggregating over long periods of time.”

I agree with this and it is supported by the analysis, but I wonder what is already known in the literature on this topic? I suspect that this is common knowledge both in the time series literature and in empirical studies using data from mobile operators for urban analytics. Please clarify if and how your findings are new considering the previous literature?

“We have also seen that there exist cells for which the different CDR layers are consistently dissimilar from one another, even though the majority of cells tend to exhibit similar behaviour across CDR layers.”

Again, is this a new finding? What do we already know regarding this? There is so much spatial analysis on CDR. How are these findings new?

We thank the reviewer for their comments on this important topic. As part of our effort in addressing the comments by both reviewers on this aspect, we have added a new paragraph in the Introduction to better place our work in the existing literature, adding further references to existing studies in the area, and we have also edited the Discussion and Conclusion section to further clarify the relationship of our results with existing studies.

17. *“For instance, this may be the case when trying to generalise a model trained on SMS data alone to one which also includes calls data.”*

Fair point, but do you have any examples of such issues? It is difficult to imagine such a fallacy.

We agree with the reviewer that this statement was potentially ambiguous, so we have removed it from the revised version of the manuscript. Whilst we believe that it is important for computational social scientists to be aware of this potential fallacy, the statement as it was did not significant information to our manuscript discussion.

Reviewer #2:

1. The paper addresses the issue of the spatio-temporal heterogeneity of mobile phone data, specifically Call Detail Records (CDRs), and the effect of the temporal and spatial aggregations when comparing the different communication/usage channels, i.e. text messages, calls and internet activity. The study is interesting and maybe a good reference point for researchers interested in computational social science, since CDRs are becoming one of the fundamental data source for the comprehension of many social issues on large scale populations. The main merit of the work is that it systematically and quantitatively address a problem which many researchers dealing with mobile phone data have tackled. CDR data are extremely heterogenous on different dimensions; not only in their spatio-temporal aspects, but also in their social aspects, i.e. how people interact through different communication channels. However, it does not represent a novelty, since many studies in the computer and mobile network community dealt with these heterogeneity - see for instance "Naboulsi, D., Fiore, M., Ribot, S., & Stanica, R. (2015). Large-scale mobile traffic analysis: a survey. *IEEE Communications Surveys & Tutorials*, 18(1), 124-161". In summary, the work may express its best potentiality for the computational social science audience.

As for the methodology, it sounds and rely on a public available dataset - even if it is bit outdate -, making easy to reproduce and verify the results presented in the paper.

We would like to thank the reviewer for their positive feedback about our manuscript, and we are pleased to hear that they agree with us in thinking that our work could be a reference point for people interested in computational social science

2. For these reasons I would suggest a minor revision referencing to previous works which have addressed and highlighted the differences in CDRs when dealing with the different layers of interaction.

Here a few suggestions:

- Naboulsi, D., Fiore, M., Ribot, S., & Stanica, R. (2015). Large-scale mobile traffic analysis: a survey. *IEEE Communications Surveys & Tutorials*, 18(1), 124-161
- Heydari, S., Roberts, S. G., Dunbar, R. I., & Saramäki, J. (2018). Multichannel social signatures and persistent features of ego networks. *Applied network science*, 3(1), 8.
- A. A. Nanavati, R. Singh, D. Chakraborty, K. Dasgupta, S. Mukherjea, G. Das, S. Gurumurthy, and A. Joshi, "Analyzing the structure and evolution of massive telecom graphs," *Knowledge and Data Engineering, IEEE Transactions on*, vol. 20, no. 5, pp. 703-718, 2008
- Zignani, M., Quadri, C., Gaito, S., & Rossi, G. P. (2015). Calling, texting, and moving: multidimensional interactions of mobile phone users. *Computational Social Networks*, 2(1), 13.

We thank the reviewer for their positive comments on our manuscript. Indeed, that is the intended audience of this manuscript. We also thank the reviewer for suggesting to improve our discussion on the relationship between our study and existing studies in the literature. In an effort to address their comment, we have added a new paragraph to the Introduction section to discuss the studies they suggested, and we have also edited the Discussion and Conclusion section to better discuss our results in relation to existing studies. Finally, we have edited the text throughout to ensure a clearer presentation of our novel results.

Appendix C

Quantifying the differences in Call Detail Records

Reviewer report

I think the authors have addressed most of my comments. But there are still a few more pending. Please see below.

The analysis of Internet activity in particular extends previous results which had already highlighted the existence of differences between mobile phone interactions data.

Please cite these studies.

The existence of spatial clustering is in line with previous studies which have shown that mobile phone interactions tend to happen more likely between individuals who are in spatial proximity [31] .

Please see the work of Arribas-Bel on LISA clustering of hourly CDR data

Arribas-Bel, D., and E. Tranos. 2018. Characterizing the spatial structure (s) of cities “on the fly”: The space-time calendar. *Geographical Analysis* 50 (2):162–181.

Future work should also look more closely at differences that arise when considering weekdays and weekends separately, as these will undoubtedly give rise to interesting behavioural differences that will be reflected in CDR data.

I do not understand what is the justification of not implementing the above in the current paper. The authors said that they “agree with the reviewer that this is a potentially very interesting direction to explore, which, however, we feel goes beyond the scope of the current study”. Why does it go beyond the scope of the study?

The authors did not answer my previous question on correlation coefficients:

Why did you choose to estimate a linear model instead of presenting the correlation coefficients?

By introducing a linear model you assume a direction of causality – even though your model is not causal. Why not just present correlation coefficients? Surely, you are not trying to say that more SMS lead to more internet usage.

Appendix D

Federico Botta
Lecturer in Data Science

College of Engineering, Mathematics and
Physical Sciences
Harrison Building
Streatham Campus
University of Exeter
North Park Road
UK, EX4 4QF

+44 (0)1392 722220
f.botta@exeter.ac.uk

17 March 2021

Revision of "Quantifying the differences in Call Detail Records"

Dear Dr Mirco Musolesi (Associate Editor) and Marta Kwiatkowska (Subject Editor),

We were extremely pleased to receive your decision letter on the 15th March 2021 with the positive feedback from the reviewer. We were glad to hear that the reviewer thought we addressed the majority of their comments.

The reviewer's final comments were indeed relevant to our manuscript, and we have further improved the quality of our manuscript as a consequence. We include below here the comments of the reviewer, as well as our responses to their points. Additionally, we also include a PDF copy of our manuscript which highlights the changes we did to our manuscript as a result of the feedback from the reviewer.

We greatly appreciate the time taken by the reviewer and by you to provide helpful and constructive feedback. We hope that our manuscript is now ready to be published in *Royal Society Open Science*.

Thank you once again for your kind help and consideration in this matter.

Sincerely,

Federico Botta

REVIEWER #1 COMMENTS

I think the authors have addressed most of my comments. But there are still a few more pending. Please see below.

“The analysis of Internet activity in particular extends previous results which had already highlighted the existence of differences between mobile phone interactions data.”

Please cite these studies.

We thank the reviewer for raising this comment. Those studies were already cited in the *Introduction* section (references 30-33) in our previous version, but we have now added a citation to those studies in the sentence highlighted by the reviewer too for completeness.

“The existence of spatial clustering is in line with previous studies which have shown that mobile phone interactions tend to happen more likely between individuals who are in spatial proximity [31].”

Please see the work of Arribas-Bel on LISA clustering of hourly CDR data

Arribas-Bel, D., and E. Tranos. 2018. Characterizing the spatial structure (s) of cities “on the fly”: The space-time calendar. *Geographical Analysis* 50 (2):162–181.

We thank the reviewer for highlighting this useful reference, which we have now added to the manuscript as suggested.

“Future work should also look more closely at differences that arise when considering weekdays and weekends separately, as these will undoubtedly give rise to interesting behavioural differences that will be reflected in CDR data.”

I do not understand what is the justification of not implementing the above in the current paper. The authors said that they “agree with the reviewer that this is a potentially very interesting direction to explore, which, however, we feel goes beyond the scope of the current study”. Why does it go beyond the scope of the study?

The focus of this manuscript is on the general relationship between different CDRs and how they change spatially and temporally. As the reviewer correctly pointed out in their first review, there is an interesting question in considering weekdays and weekends separately. However, we feel that this is part of a larger study which compares CDRs in different scenarios. For instance, in future work we intend to investigate whether there are differences not only between weekdays and weekends, but also between working days and bank holidays (such as the Christmas period included in the period of this analysis), or during different times of day (morning, afternoon or evenings), or even during specific events (such as large gatherings or protests). For such reasons, we believe that the analysis of weekends and weekdays is beyond the scope of the current study.

The authors did not answer my previous question on correlation coefficients:

Why did you choose to estimate a linear model instead of presenting the correlation coefficients?

By introducing a linear model you assume a direction of causality – even though your model is not causal. Why not just present correlation coefficients? Surely, you are not trying to say that more SMS lead to more internet usage.

We thank the reviewer for their question, and we apologise for not clearly answering this question before. In our manuscript, we present both correlation coefficients (see, for instance, Figures 2, 3C-D, and 5) as well as the results of linear regression. The relevance of the regression coefficients is in showing how the relationship between CDRs changes over time. The regression coefficient, for instance of a model including SMSs and Internet usage as variables, is effectively equivalent to the ratio of SMS activity to Internet activity. Therefore, the temporal evolution of such coefficient is informative about how the relationship between the two variables changes over time. However, the reviewer rightly points out that this gives no indication about causality, as is the case for any linear regression model since they are not designed to test causality. We have clarified in our manuscript that we do not study any causal relationship in our data set.